# A Population-Based Cross-Sectional Study of Paediatric Coeliac Disease in Catalonia Showed a Downward Trend in Prevalence Compared to the Previous Decade

**DOI:** 10.3390/nu15245100

**Published:** 2023-12-13

**Authors:** Beatriz Arau, Beatriz Dietl, Emma Sudrià-Lopez, Josefa Ribes, Laura Pareja, Teresa Marquès, Roger Garcia-Puig, Francisco Pujalte, Albert Martin-Cardona, Fernando Fernández-Bañares, Meritxell Mariné, Carme Farré, Maria Esteve

**Affiliations:** 1Digestive Diseases Department, Hospital Universitari Mútua Terrassa, Universitat de Barcelona, Pl. del Doctor Robert 5, 08221 Terrassa, Spain; beatrizarau@mutuaterrassa.es (B.A.); esudria@mutuaterrassa.cat (E.S.-L.); martincardona@gmail.com (A.M.-C.); ffbanares@mutuaterrassa.es (F.F.-B.); 2Centro de Investigación Biomédica en Red de Enfermedades Hepáticas y Digestivas (CIBERehd), Av. Monforte de Lemos 3-5, Pabellón 11, 28029 Madrid, Spain; mmarine@mutuaterrassa.cat; 3Internal Medicine and Infectious Diseases Department, Hospital Universitari Mútua Terrassa, Universitat de Barcelona, Pl. del Doctor Robert 5, 08221 Terrassa, Spain; beatriz.dietl@gmail.com; 4ICO-ICS Multicenter Hospital Tumor Registry Service, Catalan Institute of Oncology, Gran via de l’Hospitalet 199-203, 08908 L’Hospitalet de Llobregat, Spain; j.ribes@iconcologia.net (J.R.); l.pareja@iconcologia.net (L.P.); 5Department of Clinical Sciences, School of Medicine, Universitat de Barcelona, Bellvitge Campus, Carrer de la Feixa Llarga, s/n, 08907 L’Hospitalet de Llobregat, Spain; 6Cancer Epidemiology, Bellvitge Biomedical Research Institute (IDIBELL), Gran via de l’Hospitalet 199, 08908 L’Hospitalet de Llobregat, Spain; 7Nursing Department of Public Health, Mental Health and Maternal and Child, School of Medicine, Universitat de Barcelona, Bellvitge Campus, Carrer de la Feixa Llarga, s/n, 08907 L’Hospitalet de Llobregat, Spain; 8Department of Biochemistry, Hospital de Sant Joan de Déu, Passeig Sant Joan de Déu 2, 08950 Esplugues de Llobregat, Spain; tmarques@sjdhospitalbarcelona.org (T.M.); carme.farre@sjd.es (C.F.); 9Department of Paediatrics, Hospital Universitari Mútua Terrassa, Universitat de Barcelona, Pl. del Doctor Robert 5, 08221 Terrassa, Spain; rgarcia@mutuaterrassa.cat; 10Catlab, Department of Immunology, Vial St Jordi s/n, 08232 Viladecavalls, Spain; pujaltemorafrancisco@gmail.com

**Keywords:** coeliac disease (CD), prevalence, epidemiology, gluten introduction, rotavirus vaccine, breast feeding, caesarean delivery, antibiotics, infections

## Abstract

(1) Background: Previous studies showed an increased prevalence and incidence of coeliac disease (CD) over time. The objective is to ascertain whether the CD prevalence in Catalonia (a region of Southern Europe) among children aged 1–5 is as high as previously found in 2004–2009; (2) Methods: From 2013 to 2019, 3659 subjects aged 1–5 years were recruited following the previously used methodology. Factors with a potential impact on CD prevalence were investigated; (3) Results: In 2013–2019, 43/3659 subjects had positive serology, giving a standardised seroprevalence of 12.55/1000 (95% CI: 8.92; 17.40), compared to 23.62 (13.21; 39.40) in 2004–2007. The biopsy-proven crude prevalence was 7.92/1000 (95% CI: 5.50; 11.30), and the crude prevalence based on ESPGHAN criteria was 8.74/1000 (95% CI: 6.20–12.30). In contrast to 2004–2009, we did not find differences in the seroprevalence rates between 1 and 2 years vs. 3 and 4 years of age (age percentage of change −7.0 (−29.5; 22.8) vs. −45.3 (−67.5; −8.0)). Rotavirus vaccination was the most remarkable potential protective factor (48% vs. 9% in 2004–2009; *p* < 0.0001), but not the time of gluten introduction. (4) Conclusion: The present study did not confirm a worldwide CD prevalence increase and emphasizes the need to perform prevalence studies over time using the same methodology in the same geographical areas.

## 1. Introduction

Coeliac disease (CD) is an immune-mediated enteropathy triggered by gluten ingestion in genetically susceptible subjects [1]. Therefore, the prevalence of CD is influenced by the frequency of predisposing HLA haplotypes and the type and amount of cereal consumption in the general population. CD has been reported in many countries worldwide, but the prevalence varies from one region to another [2]. A meta-analysis on the prevalence of CD reported a global seroprevalence of 1.4% and biopsy-proven prevalence of 0.7%. The meta-analysis [2] and another more recent study on the incidence of CD [3] showed an increase in both prevalence and incidence over time, disclosing higher values for children than for adults and higher values for women than for men. Despite this, it is unclear whether there is any variation in CD prevalence in different areas of the world and between populations because of significant variability between studies with great methodological heterogeneity (differences in screening methods, inclusion criteria, populations) [2,3].

In the first decade of the 2000s, our group performed an observational cross-sectional survey in Catalonia (northeastern Spain) that accurately reflected population distribution in terms of sex and age (from 1 to 80 years old). This methodology was unprecedented and demonstrated a drastic significant drop in the prevalence of CD related to age in older generations (change in prevalence by age of −5% (95% CI: −7.58 to −2.42%)). This reduction was especially remarkable in the paediatric population (1–14 years), with a decrease in CD prevalence of −17% (95% CI: −25.02% to −6.10%), particularly marked in the first 4 years of life [4].

Two hypotheses were proposed to explain this unexpected finding for a disease that is lifelong: (1) The existence of an environmental cohort effect acting as a trigger in early childhood during the study period (infections, vaccines, food policies, use of antibiotics among others). In fact, a cohort effect, due to changes in the national recommendations on gluten introduction, was reported in Sweden as a “CD epidemy” [5]. (2) The appearance of tolerance to gluten with age in a proportion of cases that is more frequent among children diagnosed with CD before two years of age [6].

The aim of the present study was to ascertain whether the prevalence of CD in Catalonia in the group of children under 5 years is maintained at the same high levels found in the previous decade (cohort 2004–2007). Therefore, we designed a study with exactly the same CD screening methodology and again reproduced the reference population by age and sex in the same geographical area. In addition, factors with potential impact on CD prevalence were investigated.

## 2. Materials and Methods

### 2.1. Cross-Sectional Study (2013–2019): Subjects and Study Design

From January 2013 to December 2019, subjects aged 1–5 years were consecutively recruited at two centres, one paediatric tertiary referral hospital (Hospital Sant Joan de Déu, HSJD) and one general hospital with a paediatric department (Hospital Universitari Mútua Terrassa, HUMT). The subjects were recruited in the ambulatory minor surgery departments. Spare serum from the preoperative profile was used for CD antibody detection, avoiding a specific blood draw for this study and facilitating 100% acceptance. The predominant types of surgeries were circumcision, adenoid and tonsil surgery, ear drainage, and ophthalmological and orthopaedic surgery. In case of a positive serology, the diagnostic work-up of CD was proposed, which included confirmatory serology, genetic study and duodenal biopsy. To avoid selection bias, only individuals residing in the hospital catchment areas were included. Subject inclusion was consecutively performed by reproducing the population pyramid of Catalonia in 2013, according to data from the Catalan Statistics Institute [7]. Exclusion criteria included: heart failure or unstable cardiopathy, chronic obstructive pulmonary disease or respiratory insufficiency, coagulopathy, hepatic cirrhosis, kidney failure, active neoplasm, and gluten-free diet without an established CD diagnosis.

Subjects were classified into two age groups (1–3 years old and 3–5 years old) for each sex. Consecutive subject enrolment concluded when the previously calculated sample size in each age and sex group was achieved.

### 2.2. Antibody Detection

Serum IgA tissue transglutaminase antibody (anti-tTG2) was analysed using a quantitative automated ELISA detection kit (Elia CelikeyTM, Phadia AB, Freiburg, Germany) with recombinant human tTG as the antigen (anti-tTG2 cutoff ≥ 8 U/mL). For subjects with values between 2 and 8 U/mL, serum IgA anti-endomysial antibodies (EmA) were determined (positive titres ≥ 1/5). Total serum IgA was measured using rate nephelometry (BN II, Siemens Healthcare Diagnostics, Frankfurt, Germany). In cases of IgA deficiency, IgG-class anti-tTG2 was measured.

### 2.3. Genetic Markers

For DNA extraction, 200 μL total blood EDTA was used with the automatic extractor Qiacube using Qiamp DNA Blood Mini Kit (Qiagen, Düsseldorf, Germany). The DNA obtained was quantified by using the spectrophotometer Denovix DS-11 and the final concentration was adjusted as recommended by the commercial kit (15 ng/μL). PCR and reverse hybridization commercial kit used was the Histo Spot SSO Kits Celiac Disease (BAG Diagnostics, Lich, Germany). It is a multiplex PCR that amplifies all HLA-DQA and HLA-DQB alleles. Reverse hybridization was performed with the fully automated MR.Spot using a cup with microarrays with all the HLA-DQA-B probes. Photos of the hybridised microarrays obtained passed through an associated PC with the HISTO match software (https://www.mcdiagnostics.co.uk/histo-match-software, accessed on 5 November 2023). Samples were analysed giving the results of heterodimers in HLA genotyping (HLA-DQ2 [A1*0501/0505, B1*0201/*0202], HLA-DQ8 [A1*03, B1*0302]).

### 2.4. Duodenal Biopsy by Histopathology and Flow Cytometry

Six endoscopic biopsies from the duodenal bulb and second-third portions of the duodenum were processed using haematoxylin/eosin staining and CD3 immunophenotyping. Histopathological findings were staged according to the Marsh criteria [8], as revised by Rostami et al. [9]. We assumed that intraepithelial lymphocytosis was present when ≥25 IEL/100 epithelial cells were observed [10].

An additional biopsy from the second portion of the duodenum was taken for the assessment of lymphocyte subpopulations by flow cytometry as previously described [11].

### 2.5. Diagnosis of CD

To calculate CD prevalence, we considered the following: (1) seroprevalence, i.e., cases with positive serology; (2) biopsy-proven CD, i.e., cases with positive serology and atrophy plus cases with previous confirmed biopsy-proven CD. The diagnosis in these cases was carefully reviewed by means of positive serology and duodenal atrophy at diagnosis; (3) CD fulfilling the ESPGHAN criteria [12], establishing that a patient can be diagnosed without intestinal biopsy if they have symptoms of the CD spectrum, IgA anti-tTG2 ×10 the upper limit of normal (confirmed in a second sample) and positive EmA.

### 2.6. Ethical Considerations

This study was conducted according to the guidelines of the Declaration of Helsinki. Enrolment in this study occurred after obtaining written informed consent from the parents or legal guardians of the participating children. The study protocol was approved by the Ethics Committees of the two hospitals (HUMT PI-13/00413 and HSJD PIC-44-13; date: 26 April 2013). Researchers guaranteed strict measures for preserving children’s confidentiality.

### 2.7. Previous Cross-Sectional Study (2004–2007)

The CD prevalence of the cross-sectional study conducted from January 2004 to December 2007 was previously published [4]. In brief, it included 4230 subjects aged 1–80 years (N = 780 children aged 1–14) with an additional expanded paediatric population of 1230 children. Thus, the paediatric group consisted of 2010 children aged 1–14 years. Children aged 1–5 years (*n* = 610) were used to compare the prevalence values with those of the current study (2013–2019). The inclusion criteria, type of commercial anti-tTG2 antibody used for screening, as well as the diagnostic work-up was exactly the same as described above for the current study. Children in the historical study were recruited only from the paediatric hospital (HSJD). In Figure 1, we show the flowchart of cases included in different periods and per participating hospital.

### 2.8. Factors with Potential Impact on CD Prevalence

Using the SurveyMonkey platform (www.surveymonkey.com), an epidemiological questionnaire was carried out in the two studies to retrospectively collect risk factors associated with the development of CD. A SMS was sent to all parents of the children participating in the historical set (*n* = 610) and to those of a random sample of the current study (*n* = 600). The questionnaire was sent 3 times on different days of the week and at different times. The questionnaire included the following questions: gluten introduction before 6 months, antibiotic use before 2 years of age, hospital admission due to infection and the type of infection, caesarean delivery, breastfeeding and duration, rotavirus vaccination, and education level of parents. We registered the number of nondelivered SMSs, survey link clicks, participation acceptance and refusal, and survey completion.

### 2.9. Statistical Analyses

The sample size in the current study (N = 3659) was calculated from the CD prevalence obtained in the 1–5 age group (2.6%) of the previous study and with alpha and beta risks of 0.05 and 0.20, respectively, considering a bilateral contrast and ensuring that differences in the CD prevalence of 1% could be detected between the two studies [13]. CD prevalence rates and their 95% confidence intervals (95% CIs) in both studies were standardised by sex and age to the Catalan population on 1st January 2020 [14,15]. Prevalence rates were expressed as CD cases per 1000/children. The standardization of the CD prevalence rates according to sex and age made it possible to compare the two studies, avoiding possible biases due to differences in the population pyramids between the two periods.

Log-linear models were fitted to CD prevalence rates by age, allowing calculation of the annual percent change (APC) in prevalence. The APCs were considered statistically significant when the 95% CIs did not include 0. Negative APC values were interpreted as a decline in CD prevalence, whereas positive values showed a rise in CD prevalence [16].

The comparison of the prevalence of risk factors associated with CD between the two cross-sectional studies was performed using the chi-square test, setting the level of statistical significance at 0.05 [17]. All analyses were performed using R statistical software version 4.3.0. [18].

## 3. Results

### 3.1. CD Prevalence in Both Cross-Sectional Studies

Forty-three of the 3659 subjects (2198 males; 1461 females) had positive serology, giving a global standardised seroprevalence rate of 12.55 per 1000 children (95% CI: 8.92; 17.40) (Table 1). In Table 2, we describe the clinical characteristics, serological values, genetics, histopathological cytometric findings, and final diagnoses of cases with positive serology. Nine patients did not accept biopsy, but three of them had titres of anti-tTG2 ten times the normal limit; therefore, the biopsy-proven crude prevalence was 7.92 per 1000 (95% CI: 5.50; 11.30), and the crude prevalence based on ESPGHAN criteria was 8.74 per 1000 (95% CI: 6.20–12.30).

In Table 1, we provide a comparison of the CD standardised prevalence rates between the two age groups (1–3 years old and 3–5 years old) by hospital and period (historical and present studies). In contrast to the first study (2004–2009), in the current study (2013–2019), we did not find significant differences in the prevalence rates of CD related to age. The age-related annual percent change (APC) in CD standardised prevalence rates from 2004 to 2009 between the two groups was −45.3 (−67.5; −8.0), whereas it was −7.0 (−29.5; 22.8) in the 2013–2019 study (Figure 2). Therefore, the high CD prevalence rate found in early childhood (1–3 years) from 2004 to 2009 was not confirmed in the 2013–2019 study. Moreover, a nonsignificant trend towards the lowest CD prevalence was observed in 2013–2019 compared to the previous period (Figure 3). No differences in CD prevalence between hospitals were found (Table 1).

### 3.2. Evolution of Factors with a Potential Impact on CD Prevalence

To hypothesise the causes of the previously mentioned fluctuations in CD prevalence, we performed a retrospective assessment of factors that could potentially be involved. Table 3 shows the demographic characteristics of children whose families were invited to participate in the online survey, as well as the type of response and survey participation in both studies. There were significant differences in noncontactable participants (link not delivered or not clicked) between the current study and the historical study [121 (20.2%) vs. 309 (50.7%); *p* < 0.001], which may be attributed to less availability and lack of updated mobile phone contacts in the clinical records of the 2004–2009 participants. Importantly, no differences regarding age, sex, or CD diagnosis were found between the two groups of study survey respondents.

Regarding the CD-related environmental factors evaluated in the two studies (Table 4), we did not find differences in the time of gluten introduction (before or after 6 months), the use of antibiotics before 2 years of age, hospital admission due to infections, or the type of infection. In contrast, in the current 2013–2019 study, compared to the previous study, we found an increase in caesarean deliveries (28.2% vs. 18%; *p* = 0.018), breastfeeding (73.3% vs. 59.2% *p*=0.032) and its duration (median 24 months vs. 11 months; *p* < 0.001), and, more strikingly, a much higher proportion of rotavirus vaccination (48% vs. 9%; *p* < 0.0001). Moreover, we found an increase in the educational level of parents in 2013–2019 (56.6% with a university degree vs. 32.3%, *p* < 0.0001).

## 4. Discussion

Few epidemiological studies on CD have been performed in the same regions over time. In addition, the populations evaluated or methods to define the prevalence of CD may differ between studies, and therefore, the potentially observed differences may be mainly due to inclusion biases. For these reasons, we performed the present study to report the evolution of the prevalence of CD in the first two decades of the 2000s using exactly the same methodology in the same geographical area. The high prevalence found in early childhood (from 1 to 5 years of age) in the first decade of the 2000s motivated the present follow-up study focusing on the same stage of life [4]. Interestingly, in the present study, we did not reproduce the results found in the 2004–2007 study. The global CD prevalence from 2013 to 2019 tended to be half that in 2004 to 2007. In addition, contrary to that found in the 2004–2007 study, there were no differences between children aged 1–3 years and those aged 4–5 years. In an attempt to control all possible inclusion biases, we wanted to ascertain whether there were differences between hospitals. The paediatric hospital has an urban reference area, whereas the general hospital has a mixed rural–urban reference area, but the prevalence was identical in both centres. In addition, to avoid the influence of sex on the prevalence rates between studies, both were standardised by sex and age to the reference Catalan population in 2020.

Therefore, our study did not confirm the generally accepted concept of a progressive increase in worldwide CD prevalence [2,3]. The increased prevalence and incidence over time has also been suggested in a recent systematic review and meta-analysis focused only on paediatric population across Europe [19]. In this meta-analysis, two main types of studies were included: (1) medical record-based studies of unselected cohorts and patient case series that have reported incidence and/or prevalence of diagnosed CD and (2) prevalence studies based on screening serology of previously undiagnosed CD, with some of them adding subjects diagnosed previously from the same population. Our study belongs to the second type of design, representing the true prevalence of the general population in a predetermined period of time.

The authors of this meta-analysis emphasised the great limitations around these studies using different methodologies which are largely responsible for the great variability in prevalence values, ranging from 0.10% to 3.03% [19]. As an example, the study showing the highest seroprevalence in Europe after 2000 (3.03%) was performed in Granada (Spain) between 2009 and 2012. This study included only 198 children, which is a very low sample size when assessing a disease with a relatively low prevalence. In addition, the authors did not mention how the inclusion was conducted nor the exact setting of recruitment, and looking at the symptoms of children tested, this study is clearly a case-finding study and not a population-based prevalence study. This is the reason why the prevalence is so high [20]. Another study, included also in this meta-analysis, performed in the community of Madrid between 2004 and 2005, assessed 1291 newborns genetically tested for DQ2 in cord blood. Only 255 children aged 2–3 from 362 with positive DQ2 were serologically tested for t-TGA, identifying 15 CD patients, thus assuming a prevalence of CD in the general population of 1.1% (15/1291 children) [21]. 

The variability in the methodology is so great that considering that the prevalence of CD is increasing or decreasing if identical methodology is not used, it is very adventurous. By contrast, two prevalence studies performed in Israel, which tested blood donors who underwent serological analyses with an interval of 15 years by using the same methodology, found exactly the same prevalence over time, confirming that the rising prevalence of CD is not universal [22]. In Sweden, Ivarsson et al. compared two cohorts of children born in 1993 and 1997 and found a significantly reduced prevalence of CD in the youngest cohort, suggesting that changes in the policy of gluten introduction influenced the reduction in CD prevalence [23].

It should also be emphasised that comparisons of CD prevalence between time periods should be performed in exactly the same geographical areas. In this respect, a recent longitudinal study performed among children with a genetic predisposition for CD showed high regional variability in the cumulative incidence, suggesting that differences in environmental, genetic, and epigenetic factors strongly influence CD prevalence, even within the same country [24].

Regarding environmental factors, we performed a retrospective survey to identify potential associations with the changes in CD prevalence between the two periods. Risk factors such as age at gluten introduction, breastfeeding, caesarean delivery, infections, and exposure to antibiotics, among others, have been claimed as possible CD inducers [25]. Although the possible influence of some of these factors has been very controversial, we wanted to investigate them retrospectively once we observed differences in prevalence between the two studies. We chose variables that were easy to remember, minimising memory mistakes. For example, it was impossible to obtain information regarding the amount of gluten intake, but most parents remembered the time of gluten introduction.

We did not find any association between the time of gluten introduction, antibiotic use before 2 years of age, hospital admission due to severe infections during the first 4 years, or the types of infections that were predominantly respiratory and CD.

The rotavirus vaccine has been commercially available since 2006 in Spain and therefore was rarely administered from the period of 2004–2009, rising to almost 50% in 2013–2019. Widespread rotavirus vaccination in the more recent study suggests that it is the factor most likely to have an impact on the decreased prevalence of CD, especially considering that a herd immunity effect on the community has been demonstrated with an intermediate vaccination [26]. The protective effect of the rotavirus vaccine against CD was demonstrated in a surveillance program over more than eleven years after the end of a trial that included 19,133 children randomised to receive the RotaTeq (Kenilworth, NJ) vaccine versus placebo (1:1). The prevalence of CD was almost twice as high among placebo recipients (1.1%) than among vaccine recipients (0.6%). The study suggested that rotavirus vaccination decreases the prevalence of CD in childhood and adolescence and proposed that wild-type rotavirus may trigger CD that can be prevented or reduced by rotavirus vaccination [27].

Breastfeeding that increased in proportion and duration in the later study was considered to delay or prevent CD activation. However, a systematic review showed that there is neither a relationship between breastfeeding and the development of CD nor between the duration of breastfeeding and the appearance of CD [28], although controversy persists [29].

The percentage of caesarean deliveries increased in the 2013–2019 study compared to the previous one. Although caesarean section is known to have prolonged consequences for the intestinal microbiota, no clear association with CD prevalence has been found in previous studies [30,31]. The education level increased in the present study according to that in the general population [32]. The association between educational level and socioeconomic status and CD is still contradictory and has not been a focus of studies on screening-detected CD [33]. Therefore, the observed differences in CD prevalence might imply different rates of clinical diagnosis rather than a direct influence on CD development.

Two additional clinical aspects merit some comment. We included the assessment of lymphocyte subpopulations as complementary information since it is a useful tool for CD diagnosis in difficult cases. In fact, it is expected to find doubtful CD cases or cases with potential CD when a mass screening is applied. The ESPGHAN Guidelines 2020 suggests that an increase in % TCRγδ+ cells may help predict a possible further evolution to villous atrophy [12]. Therefore, cases 14 and 26 of the present study with Marsh 1 type lesion and increased % TCRγδ+ cells will be followed tightly because they probably have potential CD. These children were not considered coeliacs for the histological prevalence calculation in the present study. Regarding the girl, diagnosed with CD 1 year before inclusion in the study, she had a good clinical response and progressive decrease in IgA anti-tTG2, but mild levels still persisted at the time of inclusion in the study, 12 months after starting a gluten-free diet (GFD). This is not uncommon when there are very high titres before gluten exclusion, but close monitoring is obligated until negative serology is achieved.

This study has strengths and limitations. The main strength is the exact reproduction of a previously applied methodology to the present study and the use of spare serum of preoperative blood analyses that guaranteed a mass acceptance of this study. In contrast, the anxiety generated by a blood analysis in the invited children in other epidemiologic studies was one of the main reasons for participation refusal, which ranged from 15 to 20%. A limitation was the low percentage of responders to the survey and the low percentage of available mobile phones in the previous study, which limited participation. However, this limitation was counterbalanced by similar demographic characteristics between the survey participants of the two studies.

## 5. Conclusions

In summary, this study performed in Catalonia (a region of Southern Europe) showed a trend for a reduction in CD prevalence in the paediatric population, without differences among the age groups. Since wild-type rotavirus has been proposed as a trigger of CD, it is hypothesised that protective factors such as mass rotavirus vaccination of the paediatric population in our country in the last decade could account for this reduction. Future studies assessing changing trends in CD prevalence should use exactly the same methodology along the years and evaluate associations with potential protective or inducer factors. This is the only way to allow comparisons in CD prevalence in different time periods. 

## Figures and Tables

**Figure 1 nutrients-15-05100-f001:**
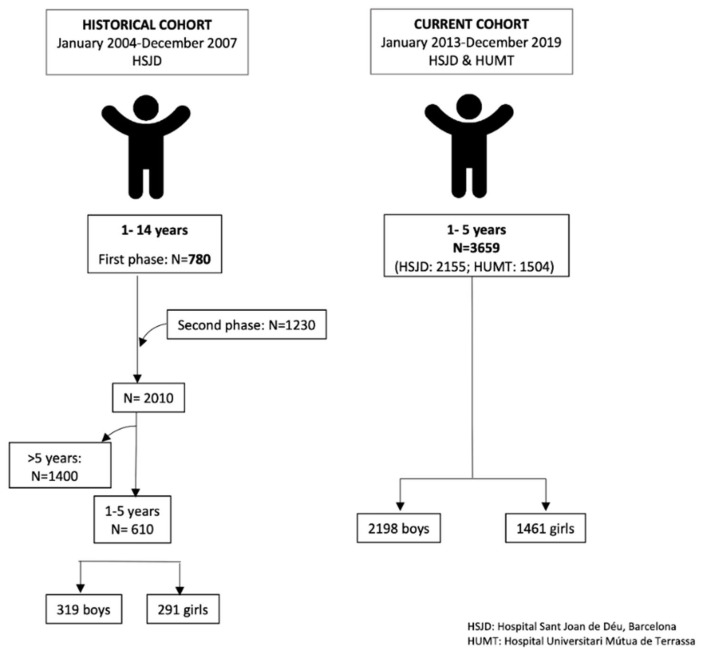
Flowchart of cases included in the historical (2004–2007) and current (2013–2019) studies.

**Figure 2 nutrients-15-05100-f002:**
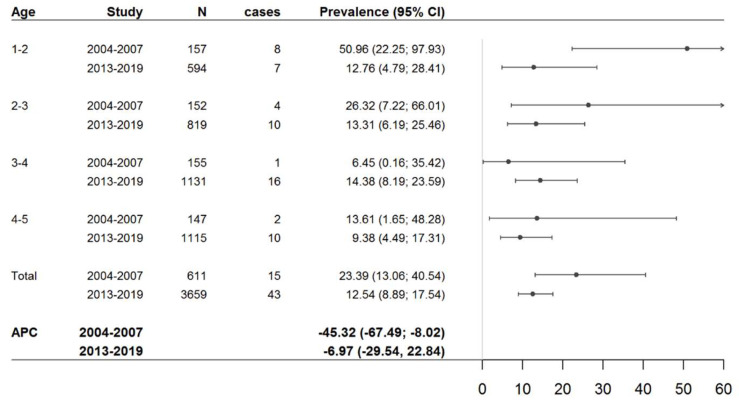
Standardised prevalence rates of coeliac disease by age and cross-sectional study period. APC: annual percent change.

**Figure 3 nutrients-15-05100-f003:**
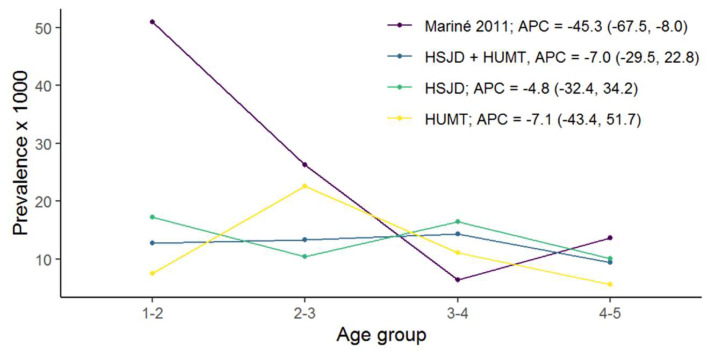
Annual percent change (APC) in coeliac disease prevalence rates by hospital and study period. HSJD: Hospital Sant Joan de Déu; HUMT: Hospital Universitari Mútua Terrassa [4].

**Table 1 nutrients-15-05100-t001:** Seroprevalence rates of coeliac disease by age group, hospital, and study period.

	Historical Study (2004–2007)			Current Study (2013–2019)								
				HSJD			HUMT			Total		
Age	CD Cases	N	SP (CI_95%_)	CD Cases	N	SSP (CI_95%_)	CD Cases	N	SP (CI_95%_)	CD Cases	N	SP (CI_95%_)
1–3	12	308	39.08 (20.19; 68.29)	12	908	12.96 (6.55; 23.54)	5	505	15.59 (4.79; 37.88)	17	1413	13.15 (7.43; 21.81)
3–5	3	302	9.81 (2.02; 29.14)	18	1247	14.06 (8.33; 22.35)	8	999	8.23 (3.37; 17.38)	26	2246	12.00 (7.82; 17.66)
Total	15	610	23.62 (13.21; 39.40)	30	2155	13.61 (9.10; 19.87)	13	1504	11.63 (5.47; 22.49)	43	3659	12.55 (8.92; 17.40)

CD: coeliac disease; SP: standardised prevalence rate. Sex- and age-standardised prevalence according to the Catalan population in 2020 (http://www.idescat.cat/pub/?id=ep&n=9123, accessed on 5 November 2023). Prevalence rates are expressed as CD cases per 1000 children.

**Table 2 nutrients-15-05100-t002:** Clinical characteristics of the patients with positive coeliac disease serology.

ID	Sex	Age (Years) *	Anti-tTG2 (IU/mL) *	Duodenal Biopsy	Cytometric Pattern	Genetic Study	Clinical Characteristics	Final Diagnosis
1	F	2.5	18	Marsh 0	Normal	DQ2.5	Asymptomatic	Non-coeliac
2	M	3.4	80	Marsh 1	ICP	DQ2.5	Asymptomatic	Non-coeliac
3	M	1.4	55	Marsh 0	Normal	DQ2.5	Asymptomatic	Giardiasis
4	M	3.9	153	Marsh 3a	CCP	DQ2.5	Asymptomatic	Coeliac
5	M	2.5	80	Marsh 3c	CCP	DQ2.5	Asymptomatic	Coeliac
6	F	3.2	10	Marsh 3b	CCP	DQ2.5	Abdominal pain	Coeliac
7	M	3.4	13	NA	NA	NA	Asymptomatic	Unknown
8 #	M	1.8	58	Marsh 3c	CCP	DQ2.5	Asymptomatic	Coeliac
9	F	3	80	Marsh 3c	CCP	DQ2.5	Asymptomatic	Coeliac
10	F	2.9	33	Marsh 3c	CCP	DQ2.5	Growth retardation	Coeliac
11 #	F	1.3	80	Marsh 3b	ICP	DQ2.5	Asymptomatic	Coeliac
12	M	4.3	80	Marsh 3c	CCP	DQ2.5	Growth retardation	Coeliac
13	F	4.3	80	Marsh 3c	CCP	DQ2.5	Asymptomatic	Coeliac
14	F	3.1	21	Marsh 1	ICP	DQ2.5	Asymptomatic	Non-coeliac
15	M	2	125	Marsh 3c	ICP	DQ2.5	Abdominal pain Iron deficiency	Coeliac
16	F	4.7	80	Marsh 3c	CCP	DQ2.5	Asymptomatic	Coeliac
17	M	2.2	52	NA	NA	NA	Asymptomatic	Unknown
18	F	1.7	103	Marsh 3c	CCP	DQ2.5	Growth retardation	Coeliac
19	M	3.2	25	Marsh 3c	CCP	DQ2.5	Iron deficiency	Coeliac
20	M	3.1	8.4	Marsh 3a	ICP	DQ2.5	Asymptomatic	Coeliac
21 #	F	1.9	80	Marsh 3c	CCP	DQ2.5	Asymptomatic	Coeliac
22	F	2.4	15	NA	NA	NA	Asymptomatic	Unknown
23	M	1.2	143	Marsh 3c	NA	DQ2.5	Growth retardation	Coeliac
24	F	3.4	102	Marsh 3b	NA	DQ2.5	Asymptomatic	Coeliac
25	F	4.7	11	Marsh 3c	CCP	DQ2.5 DQ8	Growth retardation	Previously diagnosed coeliac
26	F	2.2	15	Marsh 1	ICP	DQ2.5	Asymptomatic	Non-coeliac
27	F	4.2	28	Marsh 3c	ICP	DQ2.5	Asymptomatic	Coeliac
28	M	2.9	9.2	NA	NA	NA	Asymptomatic	Unknown
29	M	4.6	109	Marsh 3c	CCP	DQ2.5	Asymptomatic	Coeliac
30	M	3.3	156	NA	NA	DQ2.5	Growth retardation	Coeliac ^&^
31	F	2.5	80	Marsh 3c	CCP	DQ2.5	Iron deficiency	Coeliac
32	F	4.2	80	Marsh 3c	CCP	DQ8	Iron deficiency	Coeliac
33	F	4.1	56	NA	NA	NA	Asymptomatic	Unknown
34	F	3	19	Marsh 3c	CCP	DQ2.5	Asymptomatic	Coeliac
35	M	3.3	80	Marsh 3c	CCP	DQ2.5	Iron deficiency	Coeliac
36	M	1.5	16	NA	NA	NA	Asymptomatic	Unknown
37	F	3.8	80	Marsh 3b	NA	NA	Hypertransaminasemia	Coeliac
38	F	4.4	80	NA	NA	NA	Iron deficiency	Coeliac ^&^
39	F	3.1	80	Marsh 3c	CCP	DQ2.5	Iron deficiency	Coeliac
40	M	2.1	80	NA	NA	DQ2.5	Iron deficiency	Coeliac ^&^
41	F	4.8	41	Marsh 3c	CCP	DQ2.5	Asymptomatic	Coeliac
42	F	3.7	33	Marsh 3c	CCP	DQ2.5	Asymptomatic	Coeliac
43	M	3.2	59	Marsh 3b	ICP	DQ8	Iron deficiency	Coeliac

* Age at serological testing; # anti-TG2 IgA was repeated after 2 years of age. NA: not available; ICP: incomplete flow cytometric pattern (isolated increase in % TCRγδ+); CCP: complete flow cytometric pattern (increase in % TCRγδ+ (>8.5%) and a decrease in % CD3− (<10%). & ESPGHAN criteria.

**Table 3 nutrients-15-05100-t003:** Demographic characteristics of survey participants by study.

	Historical Study (2004–2007)				Current Study (2013–2019)			
	R	NR	NC	*p* Value	R	NR	NC	*p* Value
	(*n* = 99)	(*n* = 202)	(*n* = 309)		(*n* = 217)	(*n* = 262)	(*n* = 121)	
Age (Years)	N (%)	N (%)	N (%)		N (%)	N (%)	N (%)	
1–2	25 (25.3%)	59 (29.2%)	70 (22.7%)	0.513	41 (18.9%)	44 (16.8%)	20 (16.5%)	0.594
2–3	23 (23.2%)	48(23.8%)	83 (26.9%)		51 (23.5%)	53 (20.2%)	26 (21.5%)	
3–4	27 (27.3%)	54 (26.7%)	74 (23.9%)		66 (30.4%)	79 (30.2%)	30 (24.8%)	
4–5	24 (24.2%)	41 (20.3%)	82 (26.5%)		59 (27.2%)	86 (32.8%)	45 (37.2%)	
Sex								
male	54 (54.5%)	84 (41.6%)	153 (49.5%)	0.071	93 (42.9%)	115 (43.9%)	52 (43.0%)	0.971
female	45 (45.5%)	118 (58.4%)	156 (50.5%)		124 (57.1%)	147 (56.1%)	69 (57.0%)	
Coeliac disease								
yes	5 (5.1%)	2 (1.0%)	7 (2.3%)	0.087	2 (0.9%)	4 (1.5%)	1 (0.8%)	0.768
no	94 (94.9%)	200 (99.0%)	302 (97.7%)		215 (99.1%)	258 (98.5%)	120 (99.2%)	

Respondents (R): survey completed; nonrespondents (NR): survey link clicked, study participation refused, or survey not completed; noncontactable participants (NC): survey link not delivered or not clicked.

**Table 4 nutrients-15-05100-t004:** Survey results of environmental factors by study period.

	2004–2007	2013–2019	
	*n* (%)	*n* (%)	*p* Value
Gluten introduction <6 months			
Yes	33 (32.4%)	65 (28.6%)	0.479
No	58 (56.9%)	144 (63.4%)	
Unknown	11 (10.8%)	18 (7.9%)	
Antibiotic use before 2 years			
Yes	47 (47.5%)	111 (48.9%)	0.956
No	41 (41.4%)	90 (39.7%)	
Unknown	11 (11.1%)	26 (11.5%)	
Hospital admission due to infection at <5 years of age			
Yes	21 (21%)	59 (26%)	0.598
No	78 (78%)	165 (72.7%)	
Unknown	1 (1%)	3 (1.3%)	
Type of infection			
Urinary	4 (14.3%)	5 (6%)	0.342
Respiratory	15 (53.6%)	39 (46.4%)	
Enteric	0 (0%)	1 (1.2%)	
Other	7 (25%)	22 (26.2%)	
Unknown	2 (7.1%)	17 (20.2%)	
Caesarean section delivery			
Yes	18 (18%)	64 (28.2%)	0.018
No	80 (80%)	163 (71.8%)	
Unknown	2 (2%)	0 (0%)	
Breastfeeding			
Yes	58 (59.2%)	165 (73.3%)	0.032
No	34 (34.7%)	48 (21.3%)	
Unknown	6 (6.1%)	12 (5.3%)	
Months of breastfeeding duration (Median and IQR)			
	11 (6; 16)	24 (12; 36)	<0.001
Rotavirus vaccination			
Yes	9 (9%)	109 (48%)	<0.001
No	29 (29%)	73 (32.2%)	
Unknown	62 (62%)	45 (19.8%)	
Educational level of parents			
None	1 (1%)	0 (0%)	<0.001
Primary–secondary	9 (9.1%)	24 (10.6%)	
Professional	57 (57.6%)	74 (32.7%)	
University	32 (32.3%)	128 (56.6%)	

## Data Availability

The data presented in this study are available upon reasonable request from the corresponding author.

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
