# Peer review of "A Population-Based Cross-Sectional Study of Paediatric Coeliac Disease in Catalonia Showed a Downward Trend in Prevalence Compared to the Previous Decade"

_nutrients, 2023, doi:10.3390/nu15245100_

Round 1
Reviewer 1 Report
Comments and Suggestions for Authors
The authors of the manuscript addressed the important issue of assessing the incidence of coeliac disease in the youngest age group. The advantage of the study is the repetition of the research according to the same diagnostic assumptions and in similar selection of the studied population. However, it would be more appropriate to apply coeliac disease screening tests to a wider, more representative group of children from this region. Thanks to this, children without health problems and/or with various diseases, including gastrointestinal diseases, could also be included in the research.
Wouldn't it be worth referring to more epidemiological studies in the discussion, including in older children?
Similarly, based on the observed results, could it be possible to refer to the specificity of diagnostics (diagnostic tests) in celiac disease in younger children? Likewise the importance of testing lymphocyte subpopulations using flow cytometry in the diagnosis of coeliac disease in children has not been discussed.
Additionally, whether the assessment of coeliac prevalence rates by age should not refer to new cases diagnosed every year in the same group of children?
It would be worth commenting on the positive result in the girl (no. 25), who had a positive serological and genetic test results, but was previously diagnosed CD, does this mean that she did not follow a gluten-free diet after the diagnosis?
In the summary of the article, it is worth adding that the research concerned a selected region of southern Europe.
Reviewer 2 Report
Comments and Suggestions for Authors
This is quite an interesting study on the Catalonian pediatric population and this approach by continuing a previous work initially between 2004-2007. My comments can be found below:
Change the title since it focuses only on Catalonian paediatric population to be specific.
Please re-write the Abstract. In this initial form appears to be structured much more like a systematic review and meta-analysis. Reformulate the first sentence at 1) Background
Why the study has not been designed to cover 2022, maybe 2023?
Remove "etc" or use different terminology. It is not appropriate
Re-phrase the first sentence in the section 2.1. Subjects and study design
A more focused description of the 2.3. Genetic markers section is necessary. From DNA quantification to manufacturer name for reverse hybridization and not just the statement "commercial reverse hybridization"
Please expand with more than 2 sentences the Conclusion section
Does this study had exclusion criteria?
Comments on the Quality of English Language A population-based cross-sectional study of paediatric coeliac disease in Southern Europe showed a downward trend in prevalence compared to the previous decade.
